# Optimal Timing of External Ventricular Drainage after Severe Traumatic Brain Injury: A Systematic Review

**DOI:** 10.3390/jcm9061996

**Published:** 2020-06-25

**Authors:** Charlene Y. C. Chau, Saniya Mediratta, Mikel A. McKie, Barbara Gregson, Selma Tulu, Ari Ercole, Davi J. F. Solla, Wellingson S. Paiva, Peter J. Hutchinson, Angelos G. Kolias

**Affiliations:** 1Division of Neurosurgery, Department of Clinical Neurosciences, Addenbrooke’s Hospital and University of Cambridge, Cambridge CB2 0QQ, UK; yccc5@cantab.ac.uk (C.Y.C.C.); selma.tuelue@tirol-kliniken.at (S.T.); pjah2@cam.ac.uk (P.J.H.); 2Faculty of Medicine, Imperial College London, South Kensington Campus, London SW7 2AZ, UK; saniya.mediratta@gmail.com; 3Medical Research Council Biostatistics Unit, University of Cambridge School of Clinical Medicine, Cambridge CB2 0SR, UK; mikel.mckie@mrc-bsu.cam.ac.uk; 4Neurosurgical Trials Group, Newcastle University, Newcastle upon Tyne NE4 5PLE, UK; Barbara.gregson@ncl.ac.uk; 5Department of Neurosurgery, Innsbruck Medical University, 6020 Innsbruck, Austria; 6Division of Anaesthesia, University of Cambridge, Addenbrooke’s Hospital, Cambridge CB2 0QQ, UK; ae105@cam.ac.uk; 7Department of Neurology, Division of Neurosurgery, University of São Paulo, São Paulo 01246-903, Brazil; davisolla@hotmail.com (D.J.F.S.); wellingsonpaiva@yahoo.com.br (W.S.P.); 8NIHR Global Health Research Group on Neurotrauma, University of Cambridge, Cambridge CB2 0QQ, UK; 9Surgery Theme, Cambridge Clinical Trials Unit, Cambridge University Hospitals NHS Foundation Trust, Cambridge CB2 0QQ, UK

**Keywords:** neurosurgery, ventriculostomy, neurotrauma, intracranial pressure, EVD, TBI, ICP

## Abstract

External ventricular drainage (EVD) may be used for therapeutic cerebrospinal fluid (CSF) drainage to control intracranial pressure (ICP) after traumatic brain injury (TBI). However, there is currently uncertainty regarding the optimal timing for EVD insertion. This study aims to compare patient outcomes for patients with early and late EVD insertion. Following the preferred reporting items for systematic reviews and meta-analyses (PRISMA) guidelines, MEDLINE/EMBASE/Scopus/Web of Science/Cochrane Central Register of Controlled Trials were searched for published literature involving at least 10 severe TBI (sTBI) patients from their inception date to December 2019. Outcomes assessed were mortality, functional outcome, ICP control, length of stay, therapy intensity level, and complications. Twenty-one studies comprising 4542 sTBI patients with an EVD were included; 19 of the studies included patients with an early EVD, and two studies had late EVD placements. The limited number of studies, small sample sizes, imbalance in baseline characteristics between the groups and poor methodological quality have limited the scope of our analysis. We present the descriptive statistics highlighting the current conflicting data and the overall lack of reliable research into the optimal timing of EVD. There is a clear need for high quality comparisons of early vs. late EVD insertion on patient outcomes in sTBI.

## 1. Introduction

Intracranial hypertension is one of the principal secondary insults following severe traumatic brain injury (sTBI) and is related to escalating mass effect from haematomas, contusions, diffuse brain swelling, or hydrocephalus [1]. If escalating mass effect is left untreated, brain herniation and death will follow. Cerebrospinal fluid (CSF) drainage, via an external ventricular drain (EVD)—also known as ventriculostomy—is one of a number of therapies for the management of intracranial hypertension after severe traumatic brain injury (sTBI). CSF drainage has been demonstrated to be effective in reducing intracranial pressure (ICP), as well as improving cerebral perfusion pressure (CPP), cerebral oxygenation and metabolism [2]. Decreased ICP treatment intensity has also been reported with continuous CSF drainage [3]. However, there is a concern that these benefits may be attained at the cost of increased complications, with infections (most notably ventriculitis and meningitis) and catheter-related iatrogenic haemorrhage being the most common [4,5,6,7,8,9,10]. The pooled incidence for EVD-associated infection in a meta-analysis of 35 observational studies (from 1966 to 2013) is 11.4/1000 catheter-days, whilst that for placement-related haemorrhage ranges from 0 to 34.0% [8,10].

The availability of intraparenchymal ICP monitors (IPM) with fewer complications but without the capability of CSF drainage has led to ambivalence over the optimal timing of CSF drainage. The limited evidence, and thus the lack of remark within guidelines about the timing of CSF drainage in the stepwise treatment algorithm for intracranial hypertension, has contributed to substantial variation among neurocritical care units (NCCUs) internationally [11,12]. A tiered approach, with a “menu” of ICP-lowering therapies grouped together in each tier, in order of increasing complexity, has been frequently utilised in recent years [13,14]. External ventricular drainage is a first-tier treatment in the United States, Australia, and most parts of Europe [15,16,17,18], whilst in the United Kingdom and Israel, it is an optional second-tier treatment [18,19,20,21]. It was also used as a third-tier therapy in a study investigating the efficacy of CSF drainage via EVD in comparison with other treatment options [22,23].

The present systematic review aims to synthesise the evidence regarding the effectiveness and harms of EVD usage in the management of intracranial hypertension in adults with severe TBI, and to address the optimal timing for CSF drainage in clinical practice.

## 2. Methods

### 2.1. Protocol and Registration

The protocol was written using the preferred reporting items for systematic reviews and meta-analyses protocols (PRISMA)-P statement and PRISMA checklists [24,25]. It was registered in the PROSPERO database (registration number CRD42019115594) [26].

### 2.2. Ethics Approval and Consent

This systematic review and analysis did not require ethical approval.

### 2.3. Search Strategy

MEDLINE, EMBASE, The Cochrane Central Register of Controlled Trials (CENTRAL, The Cochrane Library, Latest Issue), Scopus and Web of Science were searched from their inception date to December 2019. The search strategy using MEDLINE (Ovid) is presented in Appendix A. Similar but individualised strategies were employed for the other databases. In addition, grey literature was searched on ClinicalTrials.gov and the WHO International Clinical Trials Registry Platform. Reference lists of all the retrieved papers were hand searched to avoid missing eligible studies not included in the primary search. Date of publication (post-1995 studies only), language (English studies only) and publication status restrictions were implemented.

### 2.4. Inclusion Criteria/Exclusion Criteria

A preliminary search identified a lack of head-to-head comparison of early vs. late CSF drainage. The search was broadened to include studies in the quantitative synthesis if they reported the three following components: (1) the number of patients with EVDs inserted, (2) the timing of EVD insertion, and (3) the outcomes of interest (see Section 2.7) reported specifically for the subset of patients who received an EVD. The timing of EVD (early or late) was defined either by the time of EVD insertion or the tier when the CSF drainage was initiated (Table 1). If both were reported, the timing was defined on the basis of the latter parameter. With regards to the insertion time, if an EVD is inserted at the same time as an intraparenchymal monitor (*IPM + EVD simul.*) or if it is inserted for ICP monitoring and subsequent drainage (*EVD only)*, it would be classified as “early”. If an EVD is inserted at a later stage after IPM insertion, it would be “late”. To revalidate this classification, and for studies where the time of insertion was unclear, timing was also classified according to the tier of CSF drainage. “Early” is when the CSF drainage is a first-tier ICP-lowering measure, whilst for “late”, second tier or later. Additional inclusion criteria for the population of interest included: (1) patients with sTBI (Glasgow Coma Scale (GCS) ≤8 either on admission or upon clinical deterioration); (2) >50% of the study subjects were adult patients (>16 years); (3) subjects with closed head injury; (4) subjects managed with a tiered ICP-based management protocol with use of an EVD. In studies with mixed injury types (e.g., TBI and non-traumatic subarachnoid haemorrhage), we included studies if they reported the results for our population of interest separately. Randomised controlled studies (RCTs), prospective cohort, retrospective observational cohort, and case-control studies were considered for inclusion.

Exclusion criteria were (1) subjects with penetrating or blast injuries; (2) studies with all or predominantly paediatric sTBI patients; (3) studies with <10 sTBI patients; (4) studies published before 1995, since more standardised ICP management was established after 1995; (5) conference abstracts, case reports, technical notes, letters, editorials, reviews, meta-analyses; (6) studies not published in English; (7) animal studies; (8) studies on primary decompressive craniectomy; and (9) studies on external lumbar CSF drainage.

### 2.5. Study Selection

Two authors (CYCC and SM) performed a two-step review process for the articles found from the database search using Rayyan [27]. First, the reviewers independently screened all the titles and abstracts identified from searches against the inclusion criteria. Second, the full texts of the chosen articles were retrieved and independently evaluated for eligibility according to predefined inclusion and exclusion criteria. Any discrepancies were discussed with two further reviewers (ST, AGK).

### 2.6. Data Extraction and Risk of Bias Assessment

Data from the included studies were independently extracted by two review authors (CYCC and SM) using a standardised, pre-piloted form. Data fields included: study characteristics (author, publication year, country, study design, sample size, inclusion and exclusion criteria), patient characteristics (demographics, mechanism of injury, initial GCS score, initial ICP level, other baseline differences), EVD/CSF drainage details (tier level, number of patients receiving CSF drainage, continuous or intermittent drainage strategy, volume of CSF drained), and clinical outcome. Continuous drainage was defined as EVDs kept uninterrupted in an open state, with pressure recordings performed by another ICP monitor, whereas intermittent drainage was if: (i) the EVDs were opened during ICP elevations above the thresholds defined by individual studies, or (ii) EVDs were opened regularly to drain specified volumes of CSF. Funding sources and potential conflicts of interest were also noted.

The risk of bias for the included studies was independently assessed by the two review authors (CYCC and SM), with disagreements resolved via discussion. The Cochrane Collaboration’s risk-of-bias assessment tool was employed for the RCTs, and the Risk of Bias in Non-randomized Studies—of Interventions (ROBINS-I) Cochrane Tool for non-randomised studies [28,29].

### 2.7. Outcomes

The primary outcomes were mortality at any time point and functional outcomes were assessed at 3 months or later by the Glasgow Outcome Scale (GOS) or the extended Glasgow Outcome Scale (GOS-E). Functional outcomes were dichotomised into favourable (GOS 4–5 or GOS-E 5–8) and unfavourable (GOS 1–3 or GOSE 1–4) outcome groups. Secondary outcomes were the assessment of ICP control, length of stay (LOS) in hospitals and/or intensive care units (ICU), the number of advanced ICP-lowering therapies or therapy intensity level (TIL), and EVD-related complications.

Most outcome data were dichotomous, thus the number of patients receiving CSF drainage via an EVD (*n*) and the number of outcome events of interest (E) were extracted. For continuous outcome data (e.g., hospital and ICU LOS), the mean and standard deviation (SD) were extracted.

### 2.8. Statistical Analyses

Statistical analysis was performed using R 3.6.1 and the meta 4.9-7 package [30,31]. The proportion of outcome events (E/*n*) with 95% confidence intervals were calculated for each study for mortality, unfavourable outcomes, the need for decompressive craniectomy, and visualised as forest plots further stratified by early or late EVD insertion where data were reported. More sophisticated meta-analysis taking advantage of fixed or random effects modelling was deemed inappropriate due to the limited number, size and methodological quality of the studies.

## 3. Results

### 3.1. Study Characteristics

Initial database search yielded 983 articles along with eight additional articles from other sources. Of these 991 articles, 444 articles were removed as duplicates. Then, 547 distinct articles were screened by title and abstract, following which 67 full-text articles were reviewed. Thirty-nine articles were excluded for reasons shown in Figure 1. Of note, thirteen studies were excluded due to a potential introduction of sampling bias, categorised as “biased sample” (Appendix A). This was defined either by the population of interest, or the type of intervention that the patients received. A biased population of interest, illustrated in three studies which selected a subgroup within sTBI patients, were excluded as external validity would be affected [Appendix A]. For instance, Lee (1998) only included patients with diffuse axonal injury, which is a subgroup of TBI patients that fitted a strict clinical and radiographic diagnosis [ Appendix A]. The ten remaining studies that fitted the latter definition were excluded, as these studies selectively included, or analysed the data of, patients on the basis of exposure to a higher tier intervention (e.g., craniectomy) [Appendix A]. Higher-tier ICP-lowering therapies in these studies included hyperosmolar therapy, barbiturates, and decompressive craniectomy. Twenty-eight articles met the inclusion criteria. However, seven did not report the outcomes specific to the subset of patients with an EVD. The remaining twenty-one articles had their data extracted for quantitative analysis (Table 2 and Table 3). Four relevant ongoing clinical trials were identified. (Appendix A).

A total of 4542 enrolled sTBI patients were included across the twenty-one studies (minimum 10 patients [48], maximum 2562 patients [47]), with 2746 (60.5%) patients receiving an EVD. Of these eligible studies, two were randomised controlled trials, one was a non-randomised controlled study, ten were prospective observational cohorts, and the rest were retrospective observational studies. None of the studies directly compared patients who received EVD at different timepoints (early vs. late). The study objectives varied among the studies: eight articles investigated the effect of CSF drainage via ventriculostomy on patient physiology and outcomes [2,3,33,35,42,46,48]. Ten articles focused on EVD-monitoring guided treatment, wherein EVD is used as a neuromonitoring and therapeutic tool [36,37,39,40,41,43,44,45,47,49]. The remaining three articles evaluated the effectiveness of ICP/CPP management protocols, which included EVD for the purposes of ICP monitoring and CSF drainage [32,34,38]. All except two studies had early EVD placement. Hereafter, this group is referred to as early EVD. Among these nineteen early EVD studies, EVD was predominantly used for the purposes of monitoring and drainage when the ICP or CPP exceeded a certain threshold (Table 3). One study used IPM for ICP monitoring, and EVD for continuous drainage [2]. One study had one arm receiving EVD for ICP monitoring and intermittent drainage, and the other arm receiving IPM and EVD for continuous drainage [42]. Two studies took paired IPM and EVD ICP measurements, with CSF drained according to the experimental schedules [43,48]. For the two studies with late EVD placement (i.e., late EVD group), the EVDs were solely inserted for CSF drainage via ventriculostomy as a second-line or last-tier therapy [3,22]. Sixteen studies reported at least one of the primary outcome measures [2,3,22,32,34,36,37,38,39,41,42,43,44,45,47,49]. Mortality and GOS/GOS-E were reported at various time intervals from in-hospital to 12 months. For secondary outcomes, fourteen studies recorded data on physiological parameters, which included ICP, CPP, and brain tissue oxygenation (PbtO_2_). Nine studies reported EVD-related complications. Seven studies detailed the use of advanced ICP-lowering interventions, and six studies reported the length of stay in ICU and/or hospital.

### 3.2. Primary Outcome—Mortality and GOS/GOS-E

Eight studies with 1820 patients (early EVD: *n* = 1784; late EVD: *n* = 36) reported the number of inpatient deaths (Figure 2a). Total in-hospital mortality ranges from 5 to 30% (8–30% and 5–6% for the early and late groups respectively). Eleven studies with 957 patients included GOS or GOS-E results at the end of the follow-up period. These studies were all patients in the early EVD groups. Patients were evaluated at 3 months (*n* = 3), at 6 months (*n* = 5), an average of 8.7 months (*n* = 1), and at 12 months post-injury (*n* = 2). All except two included mortality data at those time points separately, identified by GOS 1 or GOS-E 1. The two studies, instead, used a dichotomised measure of GOS (favourable and unfavourable outcomes), or reported GOS-E in three strata (GOS-E 1–2, 3–4, and 5–8) as well as the mean [2,49]. The mortality at 3-months post-injury or later ranges from 9 to 35% (Figure 2b), while 34–67% of the patients had an unfavourable outcome (Figure 3). In addition, three studies were excluded from the analysis of primary outcome, as the data did not fit our inclusion criteria. One study was excluded on the basis that the evaluation of GOS was performed at ICU discharge, since the clinicians would not be able to accurately determine whether hospitalised patients can regain independence in society [3,50]. One study reported neurological assessment at hospital discharge, dichotomised into favourable and unfavourable according to the ability to care for self [37]. It was unclear which scoring system was used, but if it had been GOS or GOS-E, it would be excluded based on the reason specified above. One study reported all-cause mortality without specifying the time frame, and thus was excluded from the analysis [40]. Due to the large confidence intervals, small sample sizes, and heterogeneity of included studies, the results were not aggregated. Whether the timing of EVD insertion affects mortality and/or functional outcomes cannot be appropriately concluded.

### 3.3. Secondary Outcomes

#### 3.3.1. ICP Control

Seven studies investigating CSF drainage recorded intracranial pressure details. All reported that the patients achieved ICP control or had reduced ICP following CSF drainage. The heterogeneity in the definitions of ICP control following EVD use prevented the aggregation of data (Table 4).

The magnitude of ICP change is provided by the mean change in ICP values before and after drainage (Definition A, Table 4). However, the timing for ICP measurements was not standardised with the definition varying substantially between the studies. The duration over which the mean ICP was calculated was also different. One measured at baseline, 1 min, 5 min, 10 min following CSF drainage to assess the dose–time interaction for ICP. One study selected four-minute periods before and after drainage to measure the ICP, whilst another study measured the ICP for 30 min after 30 min of drainage. Two studies had the mean ICP value averaged over 12 and/or 24 h before and after CSF drainage. Furthermore, the time to EVD insertion or drainage, drainage strategies and drainage volume were also heterogenous across the studies. The cumulative area under the curve of the ICP–time plots above a defined ICP threshold was also used to assess the ICP burden in one study. The number of patients with normal or raised ICP exceeding a pre-defined threshold were documented in one study. The study did not specify the specific ICP value used in defining the “sustained control of ICP” post-drainage (*n* = 14; 87.5%), or those with a “further rise of ICP” (*n* = 2) [22]. One study also investigated the effect of CSF drainage on ICP amplitude.

Six studies reported ICP data but were not specifically investigating the effect of EVD usage. Instead, the EVD usage was included as part of the ICP or CPP management protocol. We did not include the data as it cannot be ascertained whether the change in ICP is caused solely by the drainage or the cumulative effect of various ICP-lowering interventions within the protocol.

#### 3.3.2. ICP-Lowering Interventions

Seven studies, including 689 patients with an EVD (673 early, 16 late), reported the number of patients requiring decompressive craniectomy (DC) for refractory intracranial hypertension (Figure 4). Studies that did not specify whether DC was performed were excluded. A study which only mentioned surgical decompression without specifying whether it was an acute craniotomy or for the purposes of controlling intracranial hypertension was excluded due to the different prognosis of patients [40]. The early EVD group had 0–55% of patients requiring DC, whilst for the late group this was 12%.

Other advanced ICP-lowering interventions that were recorded include hyperosmolar therapy using mannitol or hypertonic saline (*n* = 2) [3,35], barbiturates (*n* = 2) [22,37], and therapeutic hypothermia (*n* = 2) [36,42].

With regards to the use of hyperosmolar therapy, Kinoshita et al. reported a lower volume of administered mannitol after CSF drainage (83.0 ± 103.0 mL vs. 625.0 ± 143.8 mL); episodes of haemodynamic instability following mannitol infusion also decreased (in terms of the number of hypotensive episodes) [35]. Lescot et al. reported a significant reduction of hypertonic saline boluses used with continuous CSF drainage when comparing the results before and after CSF initiation (mean ± SD: 0.7 ± 0.7 vs. 0.2 ± 0.6, *p* < 0.05) [3]. For barbiturates, they were used in one early and one late EVD studies for three patients in total (2/136 vs. 1/16; 1.47% vs. 6.25% respectively) after CSF drainage [22,37]. The patient who received barbiturate coma in the late EVD study by Bhargava et al. further received DC after the worsening of the ICP [22]. As for hypothermia, Griesdale et al. reported a more frequent use of hypothermia in patients with EVDs (32.7% vs. 0% for hypothermia <35 degrees Celsius; 53.1% vs. 8.2% for <38 degrees Celsius) [36]. Three patients (4.84%) received therapeutic cooling post-CSF drainage. It is worth noting that the intensity of the hypothermia may vary between the studies: one reported the number of patients who had received therapeutic hypothermia at <35 and <38 degrees Celsius [36], and the other study (Nwachuku et al.) did not specify the intensity of the hypothermia that was used [42]. Mild hyperventilation and decompressive lobectomy following persistent CSF drainage were also mentioned in one study [37]. One study mentioned a reduction in the treatment intensity in a subgroup of patients who received continuous CSF drainage, but there was no mention of which advanced ICP-lowering interventions had been employed [2].

#### 3.3.3. Length of Stay

Six studies reported the length of stay at the intensive care unit (ICU) with the mean ranging from 8.94 to 20.1 days [3,40,42,44]. Five of the six studies were in the early EVD group. Of these, three early EVD studies provided data for hospital LOS [40,44]. Further analysis was deemed to be inappropriate for this secondary outcome due to the incomplete reporting of standard deviations which would introduce bias.

#### 3.3.4. Device-Related Complications

Nine studies including 745 patients reported EVD-related infections, which were predominantly ventriculitis or meningitis. Of these, only four provided definitions which included positive culture [3], CSF biochemical parameters (white blood cells, high protein content, low glucose relative to serum) [45], positive culture plus abnormal biochemical parameters [37], as well as positive culture or CSF cell count or clinical criteria [39]. Of these, five also reported EVD-induced haemorrhage, and two technical device failures. Three EVD studies with 317 patients reported overall complications [37,40,44]. It was not possible to calculate the overall complication rate for the other six studies: the total number of individual complications were presented; combining the figures may result in some patients being accounted for more than once, thus overestimating the results. The complications described are therefore summarised in Table 5.

### 3.4. Stratified Analysis of Drainage Strategy

In the early EVD group, fourteen studies used an intermittent drainage strategy, whilst one used a continuous approach (Table 1). One study explicitly compared the intermittent and continuous drainage strategy with 31 patients in each arm; Nwachuku et al., found that continuous CSF drainage may be more effective at controlling intracranial pressure [42]. The drainage strategy was not reported in one of the studies. As for the late EVD group, one used continuous drainage, and the other study did not specify the drainage strategy. The limited studies for continuous drainage within each group (i.e., early and late, respectively), as compared to intermittent drainage, did not lend themselves to a stratified analysis as intended.

### 3.5. Risk of Bias Assessment

The overall quality of the studies was poor (Appendix A). The study by Dizdarevic et al. had a high risk of bias with regards to the blinding of trial personnel and outcome assessors [38]. The reporting of the methodological aspects of the second RCT by Kerr et al. was incomplete: the procedure of allocation concealment, and whether there was performance or detection bias through blinding, were not detailed [33]. Studies were not regarded to have selectively reported outcomes. Of the 19 observational studies, two were judged as having an overall moderate risk of bias. Fourteen were regarded to be at a serious risk of bias, with the remaining three were deemed at a critical risk of bias.

## 4. Discussion

This is the first systematic review to evaluate the influence of the timing of external ventricular drainage on important clinical outcomes in patients with severe TBI. The literature search identified no studies with a direct comparison of the two groups. We therefore intended to pool data from single-arm studies within the early and late groups. However, the pooling of data was eventually deemed inappropriate due to the high level of bias and heterogeneity amongst the included studies, as well as the imbalance of studies between the two groups. Thus, the comparison between the pooled rate as originally intended was not performed. Instead we focused on descriptive summary statistics for each individual study. In terms of primary outcome measures, the total in-hospital mortality was 26.8% and 5.56% for the early and late groups, respectively. As for the total mortality and unfavourable outcome at 3-months post-injury or later, the results were only available for the early group (26.2% and 50.3%, respectively). The comparison between these figures should not be made due to the substantial differences between the groups. As for secondary outcomes, the control of ICP and other physiological parameters were too heterogeneous to be pooled. The therapy intensity level was evaluated based on the need for decompressive craniectomy, which was 27.9% and 12.5% for the early and late groups, respectively. We were also not able to perform a stratified analysis based on the drainage strategy. Hence, no definitive conclusion can be drawn regarding the optimal timing of external ventricular drainage on the basis of the currently available literature.

Despite the lack of comparison and inferences made, this systematic review provides important insights for TBI research. First, this review identifies the absence of research on when external ventricular drainage should be instituted in the course of ICP management in TBI patients. The idea that the timing of therapies plays a role in improving outcomes is not a novel concept in TBI research. This is the subject of extensive investigation and debate for other ICP-lowering manoeuvres, such as DC or hypothermia [11]. However, there are no evidence-based recommendations within any national guidelines on this topic for external ventricular drainage. This could be due to the less serious potential complications associated with EVD placement, compared to advanced ICP-lowering interventions like DC, that physicians have to balance against the effectiveness of lowering the ICP burden. It may also be complicated by the preferred choice of ICP-monitoring modalities, as well as the clinical characteristics of the patients, for instance, the size of the ventricles or the presence of a midline shift. The recent consensus guidelines by the Brain Trauma Foundation only provided a Level III recommendation for CSF drainage in reducing ICP, as well as the superiority of continuous over intermittent drainage, based on two retrospective cohort studies [11]. A recent 2019 algorithm proposed by a consensus working group at the Seattle Severe Traumatic Brain Injury Consensus Conference (SIBICC) listed CSF drainage as a Tier 1 intervention [51]. Such variation in the timing of the intervention has also been demonstrated in a recent survey conducted in 66 neurotrauma centres, whereby 27% and 33% employed CSF drainage as a first- and second-tier therapy, respectively [18]. This reflects an urgent need for high-quality randomised controlled clinical trials to investigate this.

Our systematic database search identified more studies using EVDs earlier rather than later. Whilst this could be related to the popularity of EVDs as ICP-monitoring tools, there are therapeutic benefits. The main justifications used for early EVD insertion is that external ventricular drainage can reduce ICP and therapy intensity level, thus reduce the side-effects associated with advanced ICP-lowering interventions [2,3]. This might in turn translate into reduced mortality, improved long-term functional outcome, and a shorter stay in the ICU. However, not all patients may benefit from the earlier insertion due to inter-individual variations. Timofeev et al. reported that 54% of the patients experienced sustained ICP reduction for 24 h post-EVD insertion, which allowed the reduction of the therapeutic intensity level [2]. The remaining 46% returned to raised ICP levels either gradually or rapidly within a day of EVD insertion, despite the values being lower than pre-drainage levels. The different responses of the two groups may be attributed to CSF volume and ventricular size, which predicts the contribution of the CSF compartment to raised ICP [2,33]. The transient nature of ICP control and variation in the response to CSF drainage were also reported in another prospective study, albeit removing small volumes of CSF drainage intermittently and recording the ICP response for only 10 min [33]. In addition, the earlier insertion may subject patients to more EVD-related complications, such as infections and haemorrhages, and worsen mortality.

Second, the variability in TBI management protocols and outcome reporting, which makes the pooling of data challenging, is highlighted. Standardisation of data collection variables is crucial for comparative effectiveness analysis. Consensus-driven standardised variables have previously been proposed by a multidisciplinary working group in 2011, however, it was not adopted by all of our included studies beyond this timepoint [52]. There are two variables highlighted in this review that are worth discussing, namely the drainage strategy and ICP outcome reporting.

### 4.1. Variation in Reporting of Study Variables

#### 4.1.1. Drainage Strategy

The definition of “continuous” and “intermittent” drainage has not been well defined in existing literature. Some studies focus on the status of the EVD: whether it is in the opened or closed state. Continuous drainage is defined as an EVD kept open and ICP monitored by a separate ICP monitor [42]. Intermittent drainage is defined as opening an EVD only for ICP elevations or for a certain duration per day to drain a pre-specified volume of CSF. This definition was adopted in the review, as this concurs with our clinical perspective. Other studies put the emphasis on the purpose of the EVD [39,53]. If the focus is on ICP monitoring, then the drainage would be intermittent; whereas if the drainage was continuous, ICP monitoring would only be performed at pre-determined intervals [11]. There are also studies that did not report the method of drainage in this review. With only one retrospective study comparing the drainage strategy on patient outcomes in the adult TBI population, we propose this variable to be one of the essential data elements in TBI research, especially in studies investigating external ventricular drainage, given its potential to affect the outcome [42].

#### 4.1.2. Intracranial Pressure Outcome Reporting

Intracranial pressure control was one of the secondary outcomes assessed in this study. It is an important outcome variable, since it was on this basis that CSF drainage was added as a new topic in the recent BTF guidelines. However, this review highlights the heterogeneity in the definition of ICP control. The most common is the mean change in ICP after CSF drainage, however, the frequency of ICP measurements was not specified in some articles. This precludes comparison, since the ICP reduction post-drainage may not be sustained in some patients [2,33]. In addition, ambiguous phrases, such as the “sustained control of ICP”, were not defined properly in one of the included studies [22]. It remains unclear whether the control of ICP referred to the absence of further ICP elevations, or ICP below the threshold for triggering definitive management. There were also studies that reported the number of patients with an ICP above a certain value, which is difficult to interpret without details of the intensity of the treatment. The therapy intensity level is a scale that has been derived to overcome the limitation of a blunted ICP value as a surrogate marker from intensifying therapies [52]. Despite it being an arbitrary, it was devised by expert groups with high degrees of validity [13]. However, this was not adopted in any of the included studies.

### 4.2. Comparison with Existing Literature

This systematic review has some similarities with the studies comparing the type of ICP monitor, EVDs or intraparenchymal monitors, as both looked at the insertion of EVDs in sTBI patients. A recent systematic review comparing the effectiveness of EVD and intraparenchymal monitors (IPM) on patient outcomes used similar inclusion and exclusion criteria [54]. Nonetheless, it is important to highlight the difference between the two reviews, namely the fact that they compared EVD vs. IMP as ICP monitoring tools with a focus on ICP-monitoring-guided treatment. On the other hand, our review sought to evaluate the therapeutic use of EVDs, namely CSF drainage. A focus on the ICP monitoring device answers a separate question to a focus on CSF drainage. The impact of an ICP monitoring device on outcomes are two-fold: (1) whether EVD or IPM can give a more reliable measurement of ICP, thus better guide treatment and avoid unnecessary aggressive management; (2) any potential added advantages of the respective ICP monitors for sTBI patients. On the other hand, a focus on the therapeutic use seeks to ask the question whether the act of drainage can improve outcomes through a reduction in treatment intensity level and control ICP. However, the commonality with the two foci is EVD-related complications and their influence on patient outcomes.

### 4.3. Study Limitations

There are some limitations that need to be discussed. First, the definition of “early” and “late” is arbitrary, and one that is decided upon clinical judgement and existing practice. It was challenging to identify the tier of CSF drainage in some studies, as they only noted the pre-defined ICP threshold for initiating the drainage rather than the tier [36,41,44]. Therefore, for standardisation, we incorporated the timing of EVD insertion into the definition criteria, which was assumed to be equivalent to the tier of CSF drainage. This is concordant with clinical practice, since drainage is typically performed after ventriculostomy. However, this assumption does not always hold true, especially if EVDs are used as the preferred ICP monitoring tool. For instance, Griesdale et al. recorded that drainage is only performed if ICP exceeds 20 mmHg for more than five minutes without stimulation, but the number of patients who had ICP elevations beyond that threshold was not reported [36].

Second, due to the lack of head-on comparison between early and late EVD groups, the broadened search to identify all studies reporting EVD usage contributed to substantial heterogeneity. As such, there were differences in the inclusion criteria, interventions, drainage strategy, and outcome measures that prevented any suitable inferences or associations to be drawn. The substantial methodological and clinical diversity, and the insufficient studies in the late group, also meant that between-group comparisons were not reliable. Thus, this systematic review has not clarified the optimal timing of EVD insertion.

A third limitation relates to variations in ICP management protocols and the insufficient reporting of the number of patients who had received other concomitant therapies. The therapeutic effect of the concomitant therapies cannot be adjusted, and therefore be vulnerable to biases.

### 4.4. Deviations from the Protocol

Firstly, a qualitative analysis was not performed for the studies which did not report the outcome specifically for the subset of patients receiving EVD. Given the heterogeneity of the studies and the small number of patients who received EVD, such information was deemed to not confer any added value to the study.

Secondly, one of the primary outcomes were originally GOS or GOS-E at 6-months or later post-injury. Due to a few studies reporting GOS and GOS-E at 3 months, the outcomes at 3 months and 6 months were combined such that the primary outcome included in the analysis was changed to GOS or GOS-E at 3 months or later.

## 5. Conclusions

This systematic review highlights the lack of high-quality research into the optimal timing of external ventricular drainage in TBI management, as well as the heterogeneity in the data collection and outcome variables in EVD studies. Overall, these results need to be interpreted with caution due to the substantial heterogeneity of the included studies, which are mostly observational with serious or critical risk of bias. It identifies the need for high-quality clinical trials to investigate this issue further. Careful standardisation of other treatments and end-points will be crucial if such studies are to be definitive.

## Figures and Tables

**Figure 1 jcm-09-01996-f001:**
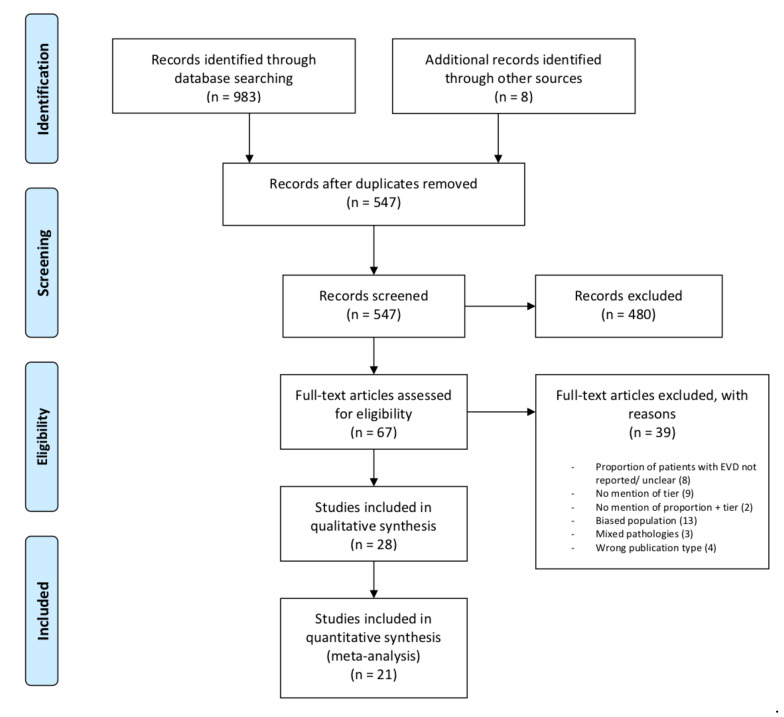
Preferred reporting items for systematic reviews and meta-analyses (PRISMA) flow diagram outlining the search strategy and reasons for the exclusion of full-text articles.

**Figure 2 jcm-09-01996-f002:**
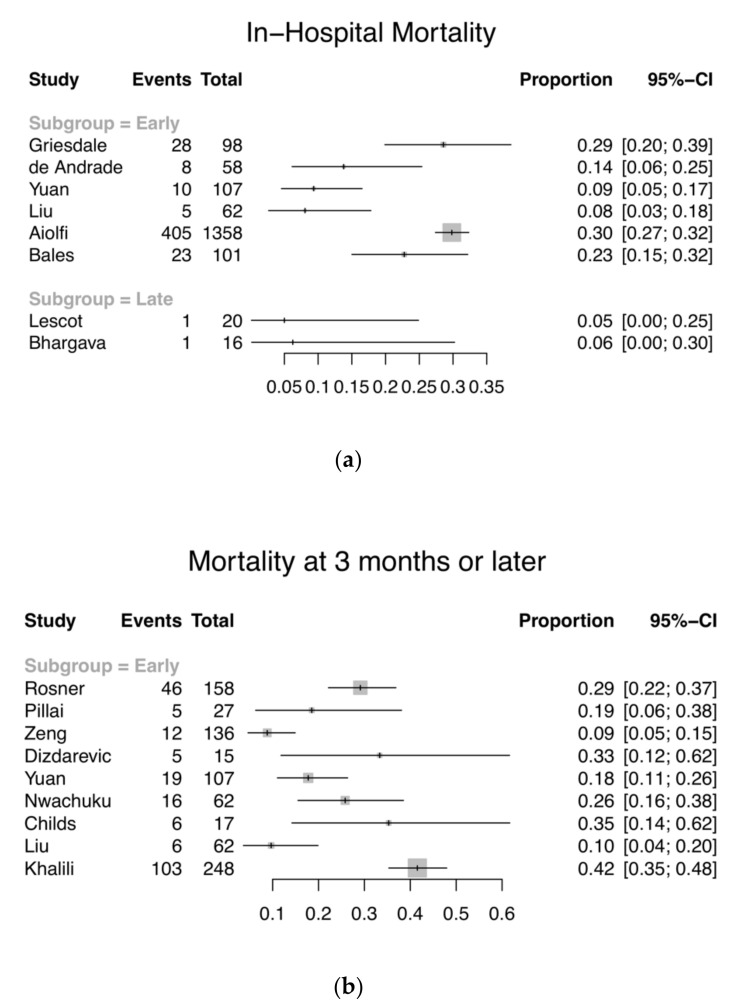
Forest plots showing the proportion and 95% confidence intervals (CI) of (**a**) in-hospital mortality; and (**b**) mortality at 3 months or later for individual studies, stratified by time of EVD insertion. (Studies in the late EVD group did not report mortality at 3 months or later).

**Figure 3 jcm-09-01996-f003:**
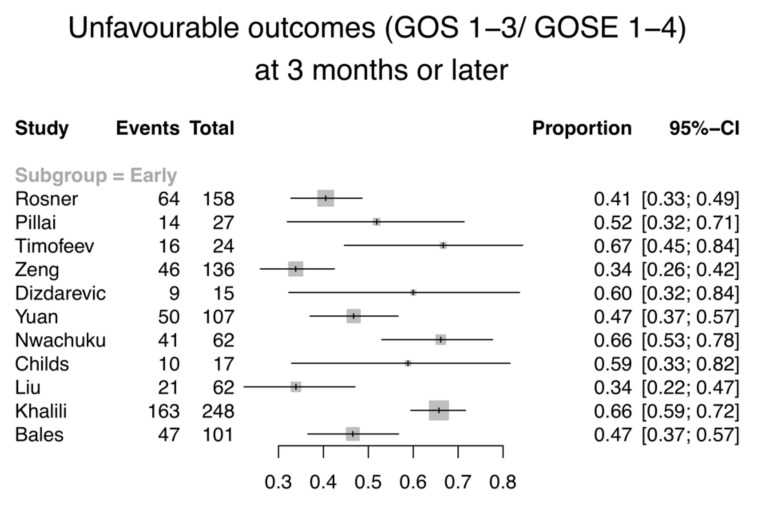
Forest plot illustrating the proportion and 95% CI of unfavourable outcomes, defined by Glasgow Outcome Scale (GOS) 1-3 or the extended Glasgow Outcome Scale (GOS-E) 1-4 at 3 months or later for individual studies. All studies are in the early EVD group. (Studies in the late EVD group did not report unfavourable outcomes at 3 months or later).

**Figure 4 jcm-09-01996-f004:**
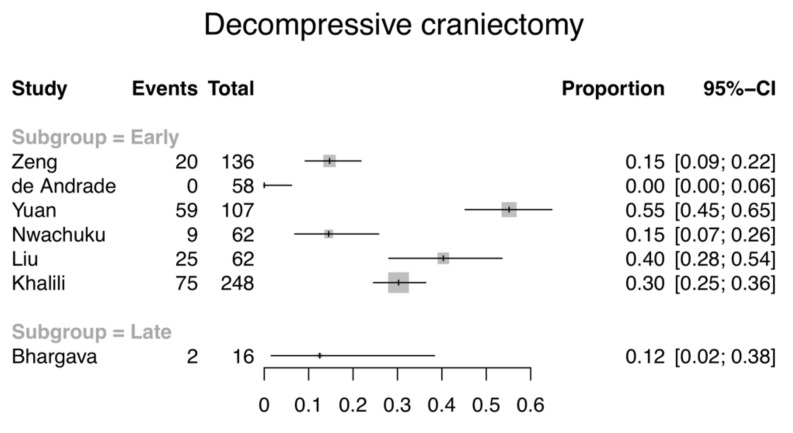
Forest plot illustrating the proportion and 95% CI of patients receiving decompressive craniectomy. All but one of the studies are in the early EVD group.

**Table 1 jcm-09-01996-t001:** Definition of the timing of EVD (early vs. late).

Defining Domains	Timing of EVD	
	Early	Late
Time of EVD Insertion	EITHER *IPM + EVD simul.*: If EVD is inserted at the same time as IPM OR *EVD only*: If EVD is inserted for ICP monitoring and subsequent drainage	Inserted at a later stage after IPM insertion
Tier/step of CSF drainage in a tiered/stepwise ICP management protocol	First tier/step	Second tier/step or later

CSF, cerebrospinal fluid; EVD, external ventricular drain; ICP, intracranial pressure; IPM, intraparenchymal monitor.

**Table 2 jcm-09-01996-t002:** Characteristics of the included studies.

Study	Study Type	Country	sTBI Patients			Mean Age	Gender (M:F)	Timing by Tier	Drainage Strategy
			Total *(N)*	EVD *(n)*	CSF *(n)*				
**Rosner, 1995 [32]**	PS	USA	158	158 (100%)	NR	27.9	117:41	Early	I
**Kerr, 2001 [33]**	RCT	USA	58	58 (100%)	58	31.6	45:13	Early	I
**Pillai, 2004 [34]**	PS	India	27	27 (100%)	25	31	NR	Early	I
**Kinoshita, 2006 [35]**	NRS	Japan	26	12 (46.2%)	12	55.3	NR	Early	I
**Timofeev, 2008 [2]**	PS	UK	24	24 (100%)	24	41	18:6	Early	C
**Griesdale, 2010 [36]**	RS	Canada	171	98 (57.3%)	NR	35	77:21	Early	I
**Zeng, 2010 [37]**	RS	China	136	136 (100%)	136	44.8	91:45	Early	I
**Dizdarevic, 2012 [38]**	RCT	BIH	15	15 (100%)	15	43	12:3	Early	I
**de Andrade, 2011 [39]**	PS	Brazil	58	58 (100%)	58	29	48:10	Early	I
**Kasotakis, 2012 [40]**	RS	USA	378	119 (31.5%)	NR	48.7	NS	Early	I
**Yuan, 2013 [41]**	PS	China	107	107 (100%)	NR	49.1	79:28	Early	I
**Nwachuku, 2014 [42]**	RS	USA	62	62 (100%)	62	34.7	42:20	Early	C (*n* = 31); I (*n* = 31)
**Childs, 2015 [43]**	PS	UK	17	17 (100%)	17	Median: 47	12:5	Early	NR
**Liu, 2015 [44]**	PS	China	62	62 (100%)	NR	41.7	50:12	Early	I
**Khalili, 2016 [45]**	PS	Iran	248	248 (100%)	NR	34.6	216:32	Early	I
**Akbik, 2017 [46]**	RS	USA	40	40 (100%)	40	39	30:10	Early	I
**Aiolfi, 2018 [47]**	RS	USA	2562	1358 (53%)	NR	Median: 52	1013:345	Early	NR
**Klein, 2018 [48]**	PS	Belgium	10	10 (100%)	10	51.9	8:2	Early	I
**Bales, 2019 [49]**	PS	USA	224	101 (45%)	86	33.6	74:27	Early	NR
**Lescot, 2012 [3]**	RS	France	20	20 (100%)	20	46.8	14:6	Late	C
**Bhargava, 2013 [22]**	RS	UK	139	16 (100%)	16	24	13:3	Late	NR

BIH, Bosnia and Herzegovina; C, continuous drainage; CSF, number of patients with CSF drained; EVD, number of patients with EVDs inserted; I, intermittent drainage; NR, not reported; NRS, non-randomised controlled study; NS, not specific to population of interest; PS, prospective observational study; RCT, randomised controlled study; RS, retrospective observational study; sTBI, severe traumatic brain injury; UK, United Kingdom; USA, United States of America.

**Table 3 jcm-09-01996-t003:** ICP monitoring and drainage details of included studies.

Study	Timing by Tier	Guidelines	ICP Monitoring	CSF Drainage Step/Tier	CSF Drainage Details
**Rosner, 1995 [32]**	Early	NR	EVD ± subdural	First step	Whenever CPP <70 mm Hg; Drain as needed: “pop-off” at 15 mmHg
**Kerr, 2001 [33]**	Early	BTF (1996)	EVD	First step (ICP >20 mm Hg)	CSF drained in random order: 1 mL (16 drops), 2 mL (32 drops), 3 mL (48 drops)
**Pillai, 2004 [34]**	Early	NR	EVD	First step (in the three-step therapeutic ladder)	NR
**Kinoshita, 2006 [35]**	Early	BTF (1996)	EVD	First step (of CPP management therapy)	NR
**Timofeev, 2008 [2]**	Early	Institutional	IPM	First tier (when ICP failed to maintain <20 mmHg and CPP >60–70 mmHg despite initial measures)	Continuous free drainage of CSF was allowed, limited only by the height of the collecting reservoir (≈15 mmHg above the external projection of foramen of Monro)
**Griesdale, 2010 [36]**	Early	Institutional	EVD	First step	If ICP >20 mmHg for >5 min without stimulation: EVD opened to 26 cm H2O; EVD closed every hour to check ICP
**Zeng, 2010 [37]**	Early	NR	EVD	First step	Monitoring with persistent intraventricular drainage; volume drained: 30–300 mL/d
**Dizdarevic, 2012 [38]**	Early	AANS (2004)	EVD	First step (when ICP >15–20 mm Hg)	NR
**de Andrade, 2011 [39]**	Early	BTF (1996)	EVD	First step	EVD kept open for 45 min with continuous drainage for 15 min if ICP overcame calibration value (10 mm Hg over foramen of Monro); EVD closed every hr to monitor ICP
**Kasotakis, 2012 [40]**	Early	NR	EVD	First step	NR
**Yuan, 2013 [41]**	Early	BTF (2007)	EVD	First tier	If ventricular pressure >20 mm Hg; Intermittent (5 min drainage) to remove the smallest volume of fluid necessary to control ICP in the shortest time
**Nwachuku, 2014 [42]**	Early	Institutional	Continuous group (*n* = 31): IPM Intermittent group (*n* = 31): EVD	First tier (when ICP > 20 mmHg for ≥5 min)	Intermittent: amount drained was variable based on individual needs to target ICP
**Childs, 2015 [43]**	Early	NR	IPM + EVD	First step	NR
**Liu, 2015 [44]**	Early	BTF (2007)	EVD	First tier	If ventricular pressure >20 mm Hg; intermittent (5 min drainage) to remove the smallest volume of fluid necessary to control ICP in the shortest time
**Khalili, 2016 [45]**	Early	Virginia stepwise ICP control	EVD	First tier	NR
**Akbik, 2017 [46]**	Early	NR	IPM + EVD	First tier	If ICP >20 mm Hg for >10 min, EVD opened to drain for 10 min and re-clamped; If ICP remains >20 mm Hg, EVD kept open at 20 cm H_2_O with ICP (IPM) recorded continuously and ICP (EVD) checked hourly
**Aiolfi, 2018 [47]**	Early	NR	EVD	First step	NR
**Klein, 2018 [48]**	Early	Institutional	IPM + EVD	First step	30 min of drainage (O1), 30 min EVD closed (C), and 30 min of drainage (O2)
**Bales, 2019 [49]**	Early	AANS	EVD	First step	NR
**Lescot, 2012 [3]**	Late	Institutional	IPM	Second-line (persistent ICP elevation > 20 mm Hg after exclusion of new surgical lesions by a repeat CT scan)	Continuous CSF drainage via EVD placed 10 cm above the external acoustic meatus.
**Bhargava, 2013 [22]**	Late	BTF (2007)	IPM	Last tier, comparing with DC/BC (definitive measures for ICP control)	NR

AANS, Americans Associations for Neurologic Surgeons; BTF, Brain Trauma Foundation; BC, barbiturate coma; CPP, cerebral perfusion pressure; CT, computed tomography; DC, decompressive craniectomy; EVD, external ventricular drain; ICP, intracranial pressure; IPM, intraparenchymal ICP monitor; NR, not reported.

**Table 4 jcm-09-01996-t004:** Four definitions of ICP control used in the seven studies that investigated the effectiveness of CSF drainage on ICP control.

Author, Year	EVD	ICP Control Description	Results
**A. Change in ICP before and after CSF drainage**			
Kerr, 2001 [33]	E; I	- Mean ICP value at baseline, 1 min, 5 min, 10 min following drainage - Decrease in ICP from baseline at various timepoints after drainage	- 1 mL CSF drained: −2.4 (1 min), −1 (10 min) mmHg * - 2 mL CSF drained: −3.4 (1 min), −1.7 (10 min) mmHg * - 3 mL CSF drained: −4.5 (1 min), −2.6 (10 min) mmHg * * values represented relative to baseline
Timofeev, 2008 [2]	E; C	Mean ICP before (≥24 h prior) and after (≥24 h) EVD	Pooled mean daily values of ICP remained <20 mmHg for at least 72 h after ventriculostomy and were significantly lower than before the procedure (*p* < 0.001).
Lescot, 2012 [3]	L; C	Mean ICP before (12 h, 24 h prior) and after (12 h, 24 h) EVD	Mean ICP before EVD: 18 ± 6 (24 h), 19 ± 7 (12 h) mmHg Mean ICP after EVD: 11 ± 5 (12 h), 12 ± 7 (24 h) mmHg Significant reduction in ICP (*p* < 0.05)
Akbik, 2017 [46]	E; I	Mean ICP change before and after EVD opening (4 min)	ICP decreased by 5.7 ± 0.6 mmHg
Klein, 2018 [48]	E; I	Mean ICP change before and after EVD (30 min)	Mean decrease after opening EVD: 2.12 ± 6.23 mmHg (*p* < 0.001)
**B. ICP burden**			
Nwachuku, 2014 [42]	E; C + I	Area under the ICP curve (amount of time with ICP > 20 mmHg)	Patients with intermittent drainage had significantly higher ICP burden than continuous drainage (59.7 ± 72.9 vs. 17.2 ± 36.8; *p* = 0.0002).
**C. ICP amplitude**			
Klein, 2018 [48]	E; I	Mean change in ICP amplitude (AMP) before and after CSF drainage	Significant reduction of amplitude of ICP signal
**D. Number of patients with normal/raised ICP values after CSF drainage**			
Bhargava, 2013 [22]	L; NR	- Number of patients with sustained control of ICP (ICP values not specified) - Number of patients with further elevation of ICP (ICP values not specified)	- Sustained control of ICP in 14 patients (87.5%) - Further elevation of ICP in 2 patients (12.5%)

C, continuous CSF drainage; E, early EVD; L, late EVD; I, intermittent CSF drainage; NR, drainage strategy not reported.

**Table 5 jcm-09-01996-t005:** EVD-related complication rates by the time of the EVD insertion.

EVD-Related Complications	Number of Patients	
	Early EVD	Late EVD
Infection	88 (12.8%)	3 (8.3%)
Haemorrhage	6 (1.5%)	NR
Device Failure	27 (14.9%)	NR
Malposition	12 (10.1%)	NR

NR, not reported.

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
