# Peer review of "Optimal Timing of External Ventricular Drainage after Severe Traumatic Brain Injury: A Systematic Review"

_jcm, 2020, doi:10.3390/jcm9061996_

Round 1
Reviewer 1 Report
The authors provide a systematic review on the timing of external ventricular drainage in severe traumatic brain injury (TBI). Following a standardized protocol for systematic reviews, several databases were searched for literature on early and late ventricular drainage in severe TBI. Of 991 articles found initially 21 fulfilled the criteria defined by the authors. 19 of them were on early intervention, 2 on late intervention. Primary outcomes were mortality and functional outcome after three months or any date later. Secondary outcomes were the assessment of ICP control, length of stay in hospitals and/or intensive care units (ICU), the number of advanced ICP-lowering therapies or therapy intensity level, and EVD-related complications. The analysis showed that the data and the studies were too heterogeneous to draw any conclusion. Therefore, the authors see the need for further studies in this field with higher quality.
Comments:
Although a definition of “early” and “late” is provided the real timing still remains unclear. Could a time interval (hours, days) be given to the reader, who is not so much into TBI?
Are patients with additional injuries in these studies excluded (50% of patients with severe TBI have additional injuries).
EVD is used to lower ICP. What is the scientific cut off for an “increased” ICP? Is the ICP really the relevant outcome parameter?
Some of the outcomes do not really provide useful information. The length of stay depends on so many variables, which are not associated with EVD (money driven, no nursing home place, additional injuries, etc.). What is the exact definition of “therapy intensity level” and how can this parameter be evaluated in a form that it can be used in a statistic (see chapter 3.3.2 which sums up additional interventions to lower ICP, which cannot really compared. Or is decompression regarded as an equivalent therapy to hypothermia or hyperosmolar therapy?
Calculating total proportions in such a heterogeneous studies is not useful, not even mortality or functional outcome. Not only the treatment is completely heterogeneous but also the patients included in the studies which is reflected by differences in the mean age of up to 27 years which also implies different trauma mechanisms (older patients: falls, younger patients traffic accidents, violence). It is confusing to report an in hospital mortality of 26.4% (early group) which is more or less the same as the three months mortality of 26.2% (due to different studies). The latter number is not only lower but it would mean that after discharge there would be no more deaths, which is completely unrealistic. It also remains unclear why there were only 5.5% deaths in the late group, which implies that the patients in these studies had less severe injuries. It might be better to eliminate all of these calculated data as they are based on studies, which are not comparable at all.
It might be better to stick with a mere description of the heterogeneous data.
Reviewer 2 Report
This article provides a thorough and well-constructed systematic review of the use of external ventricular drainage timing for the treatment of severe traumatic brain injury. It highlights, rather successfully, that current studies do not provide clear evidence to develop guidelines in the use of late or early ventricular drainage. The authors present well-articulated criticisms and comments on the current state of the literature, and address the limitations in understanding the optimal timing of intervention.
Abstract
I appreciate the inclusion of limitations and transparaency of the quality of the results in the abstract. Clear, and well written.
Introduction
A brief description about how/why increased ICP occurs following TBI would be useful. One or two sentences would be sufficient.
Overall a clear and concise introduction, making a clear case for better guidelines and design-making processes for EVD, especially given potential complications.
Methods
The methods appear to be very stringently developed, and follow best practice guidelines for systematic reviews.
Results
The figures could use some improvement for readability. The figure panel labels (.e.g (a) and (b)) are unusually placed on the right hand side. Placement on the left hand side would be better. In addition, above each panel of the forest plots a title of each forest plot would be beneficial. For example Figure 2. (a) could have the title “In-Hospital Mortality” above the first panel, and “Mortality at ≥ 3 months” for above the plot in panel (b).
I understand there is a large amount of heterogeneity in the studies, and they are of variable quality. However, it could still be valuable to have a meta-analysis of the CIs, especially for the primary outcome measures. At the very least this would demonstrate that there is little consensus or reliability in the literature. Your reasoning for other outcomes, such as ICP is very clear, given the large variability in reporting measures.
Discussion
The discussion is a comprehensive description of the weaknesses in current studies, and implementation of standardised protocols and outcome measures. A clear case is made that little can be drawn from the current literature due to hetereogeneity. This section is very well written and concise.
